# Chinese residents' environmental concern and expectation of sending children to study abroad

**Ting Yang** *, **Lejun Wang, Zihao Wang, Zeynep Safak Kutel**

School of Sociology, Huazhong University of Science and Technology, Wuhan, China

* yangting.511@163.com

## Abstract

The majority of existing studies find that Chinese residents would like to send children to study abroad for higher education quality and multiple opportunities. Previous studies have paid little attention to the association of this issue with environmental degradation in recent years. Merging data on adults from the China Family Panel Studies (CFPS) in 2016 with data on children, this paper investigates the effect of environmental concern on the educational level at which Chinese residents are willing to send children to study abroad based on the ordered logit model and Heckman Probit Model (HPM). The results show that environmental concern predicted a positive attitude toward a willingness to send children to study abroad at a decreased schooling level after concerns about the Chinese education system and educational expectations for children and other sociodemographic factors were controlled for. The marginal effects of environmental concern on expectations of sending children to study abroad at different educational levels showed that increasing environmental concern leads to the probability of residents considering sending children to study abroad during junior college or below increasing, while it leads to the probability of residents considering sending their children to study abroad during undergraduate or higher education decreasing. The HPM further verified that environmental concern had a positive effect on residents' willingness of sending their children to study during junior college or below. The study offers an important early step in the empirical testing of the relationship between Chinese residents' environmental concern and the educational level at which they would consider sending children to study abroad.

## 1. Introduction

The Ministry of Education in China announced that the number of Chinese students studying overseas soared to 662,100 in 2018, an increase of 8.83, compared to 2017 and the number of these students funded at their own expense was 467,600 [1]. Chinese Students Studying Abroad and Development (2012) published by the Center for China and Globalization (CCG) had shown that Chinese students studying abroad continued to extend from undergraduate to junior or senior high school and even lower schooling level, focusing on English-speaking

**Data Availability Statement:** The data underlying the results presented in the study are available from Institute of Social Science Survey of Peking University, China (isss.cfps@pku.edu.cn).

**Funding:** The author(s) received no specific funding for this work.

**Competing interests:** The authors have declared that no competing interests exist.

**Abbreviations:** HPM, Heckman probit model.

countries with the US and Canada [2]. In the world's major English-speaking countries, the proportion of international students from China in the basic education stage ranked first among counterparts in the same stage [2]. Take the US, the largest inflowing country of Chinese students, as an example, the monitoring data released by the US Department of Homeland Security (DHS) and the Institute of International Education (IIE) in 2017 showed that the number of Chinese students enrolled in the K12 (K12 is the collective name for basic education in the US) stage increased rapidly [3]. In a decade, the number of Chinese students enrolled in senior high schools in the US has increased by 98.6 times, far exceeding the number of postgraduates from China by 1.6 times and the number of undergraduates from China by 14 times [3]. Recent surveys showed an increasing number of Chinese parents would like to send children to study at a younger age than before. "The Survey on International Students from China" conducted by China Education Online showed that in 2015, more than two-thirds of senior high school and below students had an intention to study abroad [4]. In 2017, the "White Paper on Chinese Students Studying Abroad" published by New Oriental (China's largest service agency on studying abroad) showed that the number of junior and senior high school and below students planning to study abroad in 2016 accounted for 30% of the total students in this category, which increased by 5% in 2017 compared to 2016 [5].

Concerning the rational choice theory and the theory of planned behavior, the decision to study abroad is influenced by the potential benefits of study abroad, the constraints of resources and realization, and the assessment of norms [6]. The US, the UK, and Australia are the top three destination countries [7]. The existing literature supports the educational, social and cultural environment of these developed countries as pull factors that indicate how Chinese children are attracted by a host country. Extant studies demonstrated that the increase of Chinese students studying overseas has been guided primarily by students who seek undergraduate and master degrees [8]. There are two main reasons. First, the Chinese educational system lagged behind Western countries' educational systems, and educational concerns pushed Chinese residents to pay for their children to obtain a higher education degree abroad. Second, many students in mainland China face fierce competition and quality problems in the domestic higher education market. Many residents have adopted an independent perspective to support their children's education and are increasingly frustrated with the opportunities available in the domestic system and have turned their attention to foreign countries [7].

Environmental concern refers to awareness of high correlation between natural environment condition and life quality, such as perceived health risk of hazardous environment (e.g., air pollution, water pollution, garbage pollution). Environmental concern also implies people's behavioural intentions for improving natural environment quality. For example, people with high environmental concern would like to pay for better natural environment in the context of feasible financial burden. The rapid growth of China's economy is accompanied by environmental deterioration in the past 20 years. Environmental quality is a priority issue in China. Environmental problems such as smog have been essentially linked to basic quality of life issues such as physical health and family well-being [9]. In recent years, air pollution in China has been serious, especially in winter; it is easily noticed by residents and greatly adds to their concern about environmental degradation. With the development of rapid industrialization and motorization in recent decades, China's air pollution, especially fine particulate matter ($PM_{2.5}$) pollution has become serious, due to the continuing increase in energy consumption and the resulting multi-pollutant emissions [10]. The most pollution occurs in the east, but significant levels are considerable across northern and central China and are now not restrained to primary cities or geological basins [11]. China has adopted a countrywide air reporting system and furnished hourly statistics on air pollution, which includes airborne particulate matter (PM), $SO_2$, $NO_2$ and $O_3$ in nearly all Chinese cities [11]. Environmental concerns are a general

attitude, with a focus on cognitive and emotional evaluation of objective environmental issues. Public attention to environmental quality and sustainability is increasingly preventive as domestic environment problems continue to cause great concern. Based on postmaterialist theory [12], individuals' environmental concerns are different, and as society becomes more affluent, residents are free to achieve some postmaterialistic goals, such as environmental protection with decreasing economic struggle [13]. It is well known that poor environmental quality may damage the well-being of children [14]. Zeng, Vivian [15] analyzed 6740 Chinese children from seven cities and found an increased risk of lung function damage associated with exposure to air pollutants. Long, Lloyd [16] found that water quality and environmental sanitation were related to the frequency of diarrhea among children in a rural county of South China.

Most Chinese residents choose host countries for children such as the US, the UK, and Australia, where the natural environment is much better than that in China. It is easily speculated that environmental concern plays an ambiguous role in the decision-making process involved in children studying abroad. In the 1990s and 2000s, mainland Chinese immigrants worked hard to improve their natural and social environment and improve their quality of life for themselves and their children [17]. In addition, an increasing number of residents believe that environmental quality poses a considerable threat to the well-being of their children due to pollution or poor water quality [9]. Environmental concerns include a consumer's emotional evaluation of environmental issues and are often conceptualized as direct preconditions for environmental purchasing intentions that reflect the extent to which individuals are ready to buy products and services from companies with a good environmental reputation [14]. Just as a consumer purchases organic food at a higher price than conventionally grown food, many Chinese residents are willing to pay for their children to enjoy an improved natural environment during a relatively early education level.

The decisions to study abroad are very complicated, because these decisions are influenced by personal, economic and cultural factors, and disentangling environment-specific influences is a huge empirical challenge [14]. Although environmental concerns are usually conceptualized as a direct cause of environmental willingness to purchase, few studies have investigated the direct effects of Chinese residents' environmental concern on the educational level at which they would like to send children to study abroad mainly based on environmental intention rather than educational intention. Thus, we examine the association between environmental concern and the educational level at which their children may study abroad in a sample of Chinese residents.

## 2. Materials and methods

### 2.1. Study population

The population of the study consisted of participants from the China Family Panel Studies (CFPS), originally intended to evaluate a large proportion of China's current social phenomena. The CFPS has been implemented every two years since 2010. Using three-stage unequal probability cluster sampling, 25 provinces (Beijing, Tianjin, Shanghai, Chongqing, Hebei, Shanxi, Liaoning, Jilin, Heilongjiang, Jiangsu, Zhejiang, Anhui, Fujian, Jiangxi, Shandong, Henan, Hubei, Hunan, Guangdong, Sichuan, Guizhou, Yunnan, Shaanxi, Gansu, Guangxi) in China were covered in the survey. All participants agreed to participate, and the survey was funded by the 985 Program of Peking University and carried out by the Institute of Social Science Survey of Peking University. The participants completed the survey themselves through face-to-face or telephone interviews. The population of the study consisted of participants from the CFPS 2016. Informed consent to participate in the study was obtained from all

participants. The study was approved by the Ethical Review Committee of School of Sociology, Huazhong University of Science & Technology. Merging adult questionnaires and child questionnaires, only adults aged 18–65 years old who completed a questionnaire on demographics, environmental concerns, and expectations for their children's study abroad were selected. A total of 7464 samples were retained.

## 2.2. Variables measurement

The educational level at which children were expected to study abroad was evaluated as an ordered variable with the question, "At what level of schooling would you like to send your children to study abroad?" The answers successively included "elementary school, junior high school, senior high school, junior college, undergraduate, postgraduate, PhD". Environmental concern was measured as a continuous variable with the question, "How much do you care about the seriousness of environmental problems in China?", scored using a Likert-type 10-point scale (0 = Not at all and 10 = Very much). This single-item measurement has shown consistency across many participants. It has good face validity given that it does not link environmental concern with worldviews, behavioral intentions or attitudes, which is a limitation of alternative (longer-form) measures [18]. Educational concern was also measured by the question, "How much do you care about the seriousness of educational problems in China?" scored using a Likert-type 10-point scale (0 = Not at all and 10 = Very much). Parents' educational expectations for their children were measured by the question, "What education level would you like your children to be equipped with?" The answers included "elementary school, junior high school, senior high school, junior college, bachelor's degree, master's degree, PhD". Parents' willingness to send children to study abroad was measured with the question, "Would you consider sending your children to study abroad?" and answered by "Yes/No". The potential confounding factors selected in the models were the following categorical variables: age (≤29, 30–39, 40–49, 50–59, 60–65 years), gender, residence (rural, urban), education level (primary school or below, junior or senior high school, postsecondary or above), and neighborhood environment (good, general, poor). Statistical significance was considered for all tests at the level α = 0.05.

## 2.3. Statistical analysis

We first conducted a descriptive analysis of all participants' characteristics, demographic factors, and the educational level at which children may study abroad. Then, we ran an ordered logit model to verify the associations between residents' environmental concern and the educational level at which they would consider sending children to study abroad (seven categories) after controlling for residence, educational concerns, educational expectations and other sociodemographic factors. Next, to more precisely measure the educational level at which study abroad would be considered and consider the potential problem of endogenous sample selection, we fit an HPM for primary outcomes of environmental concern on the educational level at which they would like to send children to study abroad. The sample selection model includes two parts: a selection model and an outcome model. In the selection model, a probit model with a binary dependent variable indicates whether parents want to send their children to study abroad (i.e., equal to 1 if a parent wants to send his/her child to study abroad, 0 otherwise). To ensure that our models are well identified, we include residence, environmental concern, educational concern, and educational expectations. Therefore, assuming environmental concern, educational concerns and educational expectations is associated with willingness to send children to study abroad. The outcome model estimates a probit model in which the dependent variable is the educational level at which parents would consider sending children

to study abroad, and the independent variables include age, gender, educational level and neighborhood environment, except environmental concern, educational concerns and educational expectations.

## 3. Results

### 3.1. Characteristics of respondents

As Fig 1 shows, most of the respondents expected their children to study abroad during undergraduate education (64.03%), followed by senior high school (15.56%) or postgraduate school (9.03%). Only 1.18% of residents expected their children to study abroad during elementary school.

Table 1 lists the characteristics of the participants. The average score for environmental concern was 6.50 (range 0–10). The average age of the respondents was approximately 42 years old. Most of them were female (67.55%), had completed primary school or below (47.52%) or junior or senior high school (44.02%), and were rural residents (81.38%). In terms of the neighborhood environment, approximately 30% of the respondents reported a good environment, and over half of them reported an average environment. The mean score for educational concerns was 6.42 (range 0–10). Most parents expected that their children would obtain a college degree (66.61%), followed by a senior high school diploma (13.87%) and PhD degree (7.88%), and the fewest parents expected their children to complete only elementary school (0.37%).

Environmental concern had a significant negative effect on the educational level at which residents would consider sending children to study abroad, as shown in Table 2. Specifically, Fig 2 indicates that if the expected educational level is below 4 (junior college), the probability of considering sending children to study abroad will increase with higher environmental concern. However, if the expected educational level is 5 (college) and above, the probability of considering sending children to study abroad will decrease with higher environmental concern. Although an increase in educational concern encourages Chinese residents to consider sending children to study abroad at a higher level of schooling, the effect isn't statistically significant. Educational expectations had a positive effect on the educational level at which parents would consider sending children to study abroad. Gender, residence, educational level, age and neighborhood environment all had effects, but not statistically significant.

Furthermore, we report the results of the second stage of the HPM in Table 3, and the dependent variable in the outcome model is the educational level at which parents would consider sending children to study abroad (junior college or below vs undergraduate or above).

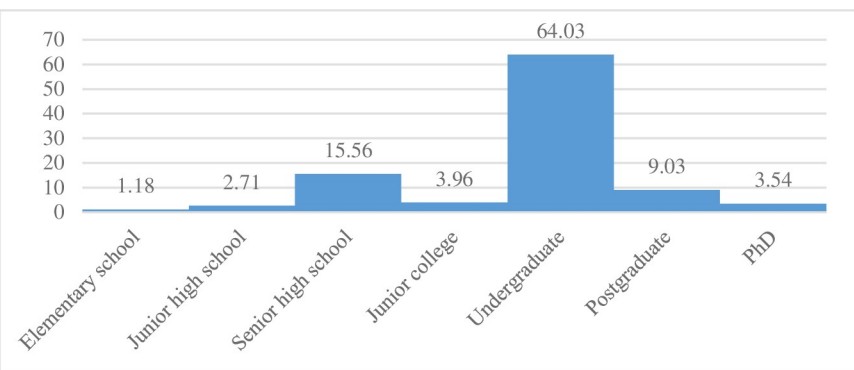

**Fig 1. Distribution of percentage of expected educational level of sending children to study abroad (%).**

**Table 1. Population characteristics.**

| Characteristics | Mean/Percent |
|---|---|
| Environmental concern | 6.50 |
| Age | 41.53 |
| Gender | |
| Female | 67.55 |
| Male | 32.45 |
| Residence | |
| Rural | 81.38 |
| Urban | 18.62 |
| Education level | |
| Primary school or below | 47.52 |
| Junior or senior high school | 44.02 |
| Postsecondary or above | 8.46 |
| Neighborhood environment(Good) | 30.43 |
| General | 53.43 |
| Poor | 16.14 |
| Educational concern | 6.13 |
| Educational expectation | |
| Elementary school | 0.37 |
| Junior high school | 2.08 |
| Senior high school | 13.87 |
| Junior college | 5.06 |
| Bachelor's degree | 66.61 |
| Master's degree | 4.11 |
| PhD | 7.88 |

The outcome model demonstrates that the coefficient of environmental concern is significant and negative, and the coefficient of the environmental concern variable in the model is 0.038 at the significance level of 1‰, which means environmental concern is associated with a higher

**Table 2. Model estimates for expected educational level of sending children to study abroad.**

| Variables | Estimate | SE | P value |
|---|---|---|---|
| Environmental concern | -0.064 | 0.023 | 0.007 |
| **Control variables** | | | |
| Educational concern | 0.003 | 0.022 | 0.879 |
| Age | 0.009 | 0.005 | 0.067 |
| Gender(female) | | | |
| Male | 0.021 | 0.118 | 0.860 |
| Residence(Rural) | | | |
| Urban | -0.034 | 0.148 | 0.820 |
| Education level(Primary school or below) | | | |
| Junior or senior high school | 0.136 | 0.125 | 0.277 |
| Postsecondary or above | 0.012 | 0.219 | 0.958 |
| Neighborhood environment(Good) | | | |
| General | 0.045 | 0.125 | 0.718 |
| Poor | 0.196 | 0.167 | 0.241 |
| Educational expectation | 0.444 | 0.055 | 0.000 |

Note: * p<0.05; ** p<0.01; *** p<0.001.

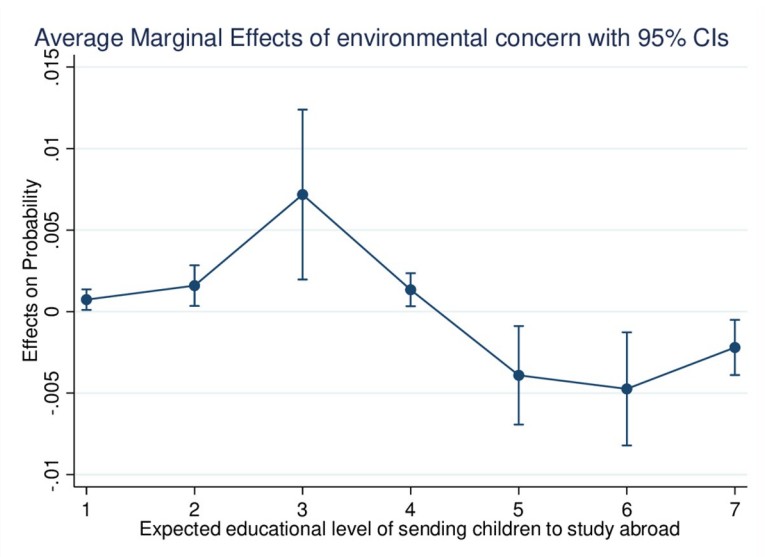

**Fig 2. Probability of the expected educational level of sending children to study abroad according to environmental concern.**

**Table 3. Heckman probit estimates for expected educational level of sending children to study abroad.**

| variables | Coefficient | Std. Err | p-value |
|---|---|---|---|
| **Outcome model** | | | |
| Environmental concern | 0.038 | 0.012 | 0.001 |
| Age | -0.004 | 0.002 | 0.082 |
| Gender(Female) | | | |
| Male | -0.041 | 0.054 | 0.450 |
| Residence(Rural) | | | |
| Urban | 0.159 | 0.073 | 0.030 |
| Education level(Primary school or below) | | | |
| Junior or senior high school | -0.098 | 0.060 | 0.102 |
| Postsecondary or above | 0.008 | 0.095 | 0.933 |
| Neighborhood environment(Good) | | | |
| General | -0.058 | 0.057 | 0.305 |
| Poor | -0.016 | 0.073 | 0.826 |
| Educational expectation | 0.041 | 0.025 | 0.105 |
| Educational concern | 0.006 | 0.011 | 0.599 |
| **Selection model** | | | |
| Environmental concern | -0.004 | 0.007 | 0.619 |
| Educational concern | 0.028 | 0.007 | 0.000 |
| Educational expectation | 0.234 | 0.017 | 0.000 |
| Residence(rural) | | | |
| urban | 0.149 | 0.042 | 0.000 |
| Athrtho | 2.571 | 1.509 | 0.088 |
| *P-value* | | 0.000 | |
| Observations | | 7464 | |
| Selected samples | | 1440 | |

LR test of rho = 0, p = 0.0018.

***p<0.001, **p<0.01, *p<0.05.

probability of expecting children to study abroad during junior college or below (vs under-graduate or higher). Compared with rural residents, urban residents are more likely to expect their children to study abroad during senior high school or below. The selection model shows that urban residence, educational concern, and educational expectations are linked to a higher probability of expecting children to study abroad. Compared with rural residents, urban residents are more likely to expect their children to study abroad. Educational concerns played a positive role in promoting residents' willingness to consider sending children to study abroad, as did educational expectations. It is particularly important to highlight that the *p*-value of the LR test of rho is 0.0018 <0.01, which demonstrates that a willingness to consider sending children to study abroad is associated with the educational level at which parents would consider sending to study abroad and that the HPM is appropriate.

## 4. Discussion

Approximately 64% of Chinese residents are willing to consider sending children to study abroad at the college level based on the CFPS 2016. Given the international mobility of students in higher education, a university's prestige is the most important factor leading Chinese residents to consider sending children to study abroad, in addition to the broad social and cultural services offered by a university [8]. Undergraduates are the main group of Chinese parents sending children to study abroad [1]. The main reason for the increase in Chinese students studying abroad during undergraduate education is the increase in the number of students seeking undergraduate and master's degrees [8]. Individuals seek the ideal university to enhance their labor market opportunities and career positioning for a future life overseas. Overseas higher education can bring a wealth of employment prospects for children when they return to China because in an increasingly competitive domestic labor market, having a degree from a prestigious overseas institution helps individuals gain a competitive advantage compared to their peers [19].

While controlling the variables that influence the willingness to consider sending children to study abroad, such as educational concerns (about the Chinese educational system), socio-economic factors (educational level, community environment), and demographic factors (age, gender, residence), the ordered logit results indicated that residents were more likely to consider sending children to study abroad at a lower level of schooling level when they had higher environmental concerns. Specifically, the marginal effects of environmental concern on Chinese residents' expectations of studying abroad at different levels of education showed that higher environmental concern tends to increase the willingness to consider sending children to study abroad at the level of junior college or below and to reduce the willingness to consider sending children to study abroad during college or higher education. The HPM further verified this by specifying how environmental concern impacts the educational level at which Chinese residents would consider sending children to study abroad. Environmental quality is the main factor affecting individuals' environmental concern [20]. It is assumed that parents are responsible to some extent for some unhealthy consequences brought about by environmental problems. They believe that their children would benefit from living and studying in a better environment in the future. They see international education as an opportunity to achieve global academic standards and achieve child migration goals.

Educational concerns and educational expectations had a positive impact on Chinese residents' willingness to consider sending children to study abroad, while they had no significant impact on the educational level at which residents would consider sending children to study abroad. While environmental concern had no significant impact on the willingness to consider sending children to study abroad, it had a positive impact on residents' willingness to consider

sending children to study abroad during junior college or below (vs. undergraduate or higher education). Educational factors lead Chinese residents to expect their children to enjoy higher-quality education in Western countries. Extant research in the context of Chinese students studying overseas has revealed push factors that include the value of obtaining a foreign degree and the high level of competition for obtaining a domestic university degree. Regarding the quality of higher education, the driving factor is that institutions in other countries provide better learning conditions than domestic providers [21, 22]. Environmental factors push Chinese residents to send their children to study abroad at a lower level of education and adapt to the natural and social environment abroad at a young age. This case differs from the international mobility of Chinese students seeking higher education.

This study has some limitations. The study did not control the income level of respondents when selecting those interested in considering sending children to study abroad and discussing their environmental concerns and the educational level at which they would consider sending children to study abroad. This implied that low-income residents must continue to pay attention to their own survival and material problems, while the rich may have a higher environmental concern and the ability to pay for their children's study abroad. However, income level and individuals' environmental concern is still a debatable point. Although the rich can protect themselves more effectively from environmental pollution than the poor can through private investment, the poor are more vulnerable to environmental pollution, which may increase their concern for the environment and their willingness to invest in a healthy environment for their children at all costs [23]. The study did not discuss whether the individuals who preferred that their children study abroad at a lower level of schooling expected them to continue to study and live abroad or even migrate. Understanding this factor may further verify that sending children to study abroad at a lower level of schooling can be the first step parents take to achieve their children's migration.

## 5. Conclusion

Merging adult data with child data, the paper explores the effects of environmental concern on the educational level at which Chinese residents consider sending children to study abroad by using ordered logit and HPM. The results show that environmental concern has a significant positive effect on sending children to study abroad during junior college or below (vs undergraduate or higher education). After controlling for educational factors and individual characteristics, environmental concern has an important impact on the educational level at which individuals consider sending children to study abroad.

The above findings are of great significance. First, since an increasing number of Chinese residents are sending their children to study abroad at a young age, the government should not only closely follow educational quality but also pay more attention to residents' demand for higher environmental quality. Second, Chinese residents pay increasing attention to their own health and the health of their children, and studying abroad at a young age may be one of the strategies many Chinese residents pursue to help their children migrate, considering that economic development enables more Chinese residents to pay for study abroad. Finally, the strategy of sacrificing the environment to develop the economy is inappropriate, and efforts should be made to balance environmental protection and economic development in China.

## Supporting information

**S1 Dataset.**
(DTA)

## Acknowledgments

We would like to thank the data support provided by the Institute of Social Science Survey of Peking University and adult participants for their voluntary contributions to the study.

## Author Contributions

**Conceptualization:** Ting Yang.

**Formal analysis:** Ting Yang.

**Methodology:** Lejun Wang.

**Software:** Zihao Wang, Zeynep Safak Kutel.

**Writing – original draft:** Ting Yang.

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
