## [Decision Letter · Decision Letter 0]

15 May 2020

PONE-D-19-29398

Chinese residents'environmental concern and expected educational level to send children to study abroad

PLOS ONE

Dear Dr Yang,

Thank you for submitting your manuscript to PLOS ONE. After careful consideration, we feel that it has merit but does not fully meet PLOS ONE’s publication criteria as it currently stands. Therefore, we invite you to submit a revised version of the manuscript that addresses the points raised during the review process.

In addition to the reported comments, the reviewer adds:

"The study is well executed because the hypotheses were clearly stated, the data collection and analyses were appropriate, and the discussion of findings did not go beyond the study results. That is, the study followed appropriate protocols for such a study.

In regard to the issue of "environmental concerns" it would be useful to have the authors spell out in more detail what types of environmental concerns they are looking at. This is a very broad statement and the reader would be helped if the types of environmental concerns of interest to the authors were described in a few more sentences."

 In addition, I strongly recommend that you revise the english fo your manuscript with the help of a native Enhlish proofreader.

We would appreciate receiving your revised manuscript by Jun 29 2020 11:59PM. To enhance the reproducibility of your results, we recommend that if applicable you deposit your laboratory protocols in protocols.io, where a protocol can be assigned its own identifier (DOI) such that it can be cited independently in the future. For instructions see: http://journals.plos.org/plosone/s/submission-guidelines#loc-laboratory-protocols

We look forward to receiving your revised manuscript.

Kind regards,

Nicola Lacetera

Academic Editor

PLOS ONE

Journal Requirements:

5. Your ethics statement must appear in the Methods section of your manuscript. If your ethics statement is written in any section besides the Methods, please move it to the Methods section and delete it from any other section. Please also ensure that your ethics statement is included in your manuscript, as the ethics section of your online submission will not be published alongside your manuscript.

Reviewers' comments:

Reviewer's Responses to Questions

**Comments to the Author**

1. Is the manuscript technically sound, and do the data support the conclusions?

Reviewer #1: Yes

2. Has the statistical analysis been performed appropriately and rigorously? 

Reviewer #1: Yes

3. Have the authors made all data underlying the findings in their manuscript fully available?

Reviewer #1: Yes

4. Is the manuscript presented in an intelligible fashion and written in standard English?

Reviewer #1: Yes

5. Review Comments to the Author

Reviewer #1: Paper discusses an interesting connection possible between desires for children to study abroad and concerns about environment in China. While these two variables do not have immediate linkage, it is interesting to speculate about this connection and the study seems to have been done carefully using existing data from a survey. The topic would be of interest to many readers. One addition that would be valuable would be more information about the questions on "environmental concerns" as this is a very broad topic.

6. PLOS authors have the option to publish the peer review history of their article (what does this mean?). If published, this will include your full peer review and any attached files.

Reviewer #1: No

---

## [Author Response · Author response to Decision Letter 0]

30 Jul 2020

Reviewer:

Paper discusses an interesting connection possible between desires for children to study abroad and concerns about environment in China. While these two variables do not have immediate linkage, it is interesting to speculate about this connection and the study seems to have been done carefully using existing data from a survey. The topic would be of interest to many readers. One addition that would be valuable would be more information about the questions on "environmental concerns" as this is a very broad topic.

Reply: Dear Reviewer, thank you very much for the valuable advice. The following are my revision based on your advice. 

The definition of environmental concern in the study has been presented and specified, as shown:

“Environmental concern refers to awareness of high correlation between natural environment condition and life quality, such as perceived health risk of hazardous environment (e.g., air pollution, water pollution, garbage pollution). Environmental concern also implies people’s behavioural intentions for improving natural environment quality. For example, people with high environmental concern would like to pay for better natural environment in the context of feasible financial burden.” (Line 5, P5)

Editor:

1."The study is well executed because the hypotheses were clearly stated, the data collection and analyses were appropriate, and the discussion of findings did not go beyond the study results. That is, the study followed appropriate protocols for such a study.

In regard to the issue of "environmental concerns" it would be useful to have the authors spell out in more detail what types of environmental concerns they are looking at. This is a very broad statement and the reader would be helped if the types of environmental concerns of interest to the authors were described in a few more sentences."

Reply: Dear Editor, thank you very much for the generous help. The following are my revision based on your advice.

As shown in the Line5, P5, “Environmental concern refers to awareness of high correlation between natural environment condition and life quality, such as perceived health risk of hazardous environment (e.g., air pollution, water pollution, garbage pollution). Environmental concern also implies people’s behavioural intentions for improving natural environment quality. For example, people with high environmental concern would like to pay for better natural environment in the context of feasible financial burden.” These sentences specify the implication of environmental concern.

2.In addition, I strongly recommend that you revise the english for your manuscript with the help of a native English proofreader.

Reply: Dear Editor, thanks for the kind reminding. I have invited AJE to proofread the manuscript as shown in the revised manuscript.

---

## [Decision Letter · Decision Letter 1]

12 Aug 2020

Chinese residents'environmental concern and expectation of sending children to study abroad

PONE-D-19-29398R1

Dear Dr. Yang,

We’re pleased to inform you that your manuscript has been judged scientifically suitable for publication and will be formally accepted for publication once it meets all outstanding technical requirements.

Kind regards,

Nicola Lacetera

Academic Editor

PLOS ONE

Additional Editor Comments (optional):

Reviewers' comments:

Reviewer's Responses to Questions

**Comments to the Author**

1. If the authors have adequately addressed your comments raised in a previous round of review and you feel that this manuscript is now acceptable for publication, you may indicate that here to bypass the “Comments to the Author” section, enter your conflict of interest statement in the “Confidential to Editor” section, and submit your "Accept" recommendation.

Reviewer #1: All comments have been addressed

2. Is the manuscript technically sound, and do the data support the conclusions?

Reviewer #1: (No Response)

3. Has the statistical analysis been performed appropriately and rigorously? 

Reviewer #1: (No Response)

4. Have the authors made all data underlying the findings in their manuscript fully available?

Reviewer #1: (No Response)

5. Is the manuscript presented in an intelligible fashion and written in standard English?

Reviewer #1: (No Response)

6. Review Comments to the Author

Reviewer #1: (No Response)

7. PLOS authors have the option to publish the peer review history of their article (what does this mean?). If published, this will include your full peer review and any attached files.

Reviewer #1: No

---

## [Editor Report · Acceptance letter]

14 Sep 2020

PONE-D-19-29398R1 

Chinese residents'environmental concern and expectation of sending children to study abroad 

Dear Dr. Yang:

I'm pleased to inform you that your manuscript has been deemed suitable for publication in PLOS ONE. Congratulations! Your manuscript is now with our production department. 

Kind regards, 

on behalf of

Professor Nicola Lacetera 

Academic Editor

PLOS ONE